# Antiproliferative Imidazo-Pyrazole-Based Hydrogel: A Promising Approach for the Development of New Treatments for PLX-Resistant Melanoma

**DOI:** 10.3390/pharmaceutics15102425

**Published:** 2023-10-04

**Authors:** Silvana Alfei, Marco Milanese, Chiara Brullo, Giulia Elda Valenti, Cinzia Domenicotti, Eleonora Russo, Barbara Marengo

**Affiliations:** 1Section of Chemistry and Pharmaceutical and Food Technologies, Department of Pharmacy, University of Genoa, Viale Cembrano, 4, 16148 Genoa, Italy; marco.milanese@unige.it; 2Section of Medicinal Chemistry and Cosmetic Product, Department of Pharmacy (DIFAR), University of Genoa, Viale Benedetto XV, 3, 16132 Genoa, Italy; chiara.brullo@unige.it (C.B.); eleonora.russo@unige.it (E.R.); 3Department of Experimental Medicine (DIMES), University of Genova, Via Alberti L.B., 16132 Genoa, Italy; giuliaelda.valenti@edu.unige.it (G.E.V.); cinzia.domenicotti@unige.it (C.D.)

**Keywords:** imidazo-pyrazoles (IPs), PLX-resistant melanoma, cationic polystyrene-based resin (R4), self-formed composite hydrogel, high equilibrium degree of swelling (EDS), high equilibrium water content (EWC), shear thinning, pseudoplastic rheological behavior, low yield stress

## Abstract

Aiming at developing a dermal formulation against melanoma, the synthesized imidazo-pyrazoles 2-phenyl-2,3-dihydro-1H-imidazo[1,2-b]pyrazole-7-carboxylic acid (3-methoxy-4-phenoxy-benzylidene)-hydrazide (**4G**) and 2-phenyl-2,3-dihydro-1H-imidazo[1,2-b]pyrazole-7-carboxylic acid (4-benzyloxy-3-methoxy-benzylidene)-hydrazide (**4I**) were screened on patient-isolated melanoma cells (MEOV NT) and on Vemurafenib (PLX4032)-resistant (MEOV PLX-R) ones. Since **4I** on MEOV PLX-R cells was 1.4-fold more effective than PLX, a hydrogel formulation containing **4I** (R4HG-4I) was prepared in parallel with an empty R4-based hydrogel (R4HG) using a synthesized antibacterial resin (R4) as gelling agent. Thanks to its high hydrophilicity, porosity (85%), and excellent swelling capability (552%), R4 allowed to achieve R4HG and R4HG-4I with high equilibrium degree of swelling (EDS) and equilibrium water content (EWC). Chemometric-assisted ATR-FTIR analyses confirmed the chemical structure of swollen and fully dried (R4HG-D and R4HG-4I-D) hydrogels. The morphology of R4HG-D and R4HG-4I-D was examined by optical microscopy and SEM, while UV–vis analyses were carried out to obtain the drug loading (DL%) and the encapsulation efficiency (EE%) of R4HG-4I. Potentiometric titrations were performed to determine the equivalents of NH_3_^+^ in both R4HG and R4HG-4I. The swelling and water release profiles of both materials and related kinetics were assessed by equilibrium swelling rate and water loss studies, respectively, while their biodegradability over time was assessed by in vitro degradation experiments determining their mass loss. Rheological experiments established that both R4HG and R4HG-4I are shear-thinning Bingham pseudoplastic fluids with low yield stress, thus assuring easy spreadability in a future topical application. Release studies evidenced a sustained and quantitative release of **4I** governed mainly by diffusion. Upon favorable results from further experiments in a more realistic 3D model of melanoma, R4HG-4I could represent a starting point to develop new topical therapeutic options to adjuvate the treatments of melanoma cells also when resistant to currently available drugs.

## 1. Introduction

Human skin cancer is currently considered as the most common cancer diagnosed, especially in Caucasian people worldwide [1]. It comprises melanoma skin cancer (MSC) and non-melanoma skin cancer (NMSC) [2], which in turn comprises basal cell carcinoma (BCC) and squamous cell carcinoma (SCC) [3]. Malignant melanoma (MM) is a human SC derived from transformed melanocytes, whose incidence continues to increase worldwide [4]. By rapid widespreading metastasis to the lymphatic system and other important organs, melanoma accounts for approximately 73% of skin-cancer-related deaths [5,6]. There are several risk factors, such as increased occupational and recreational UV light exposure, contact with certain chemicals, as well as smoking [7] and prepuberty sunburn. Additionally, skin infections sustained by opportunistic bacteria, favored by a relatively weak immune system of the skin tissue, can aggravate further the intricacy of MM, thus causing ulcerations and improving the mortality of patients [8]. Current standard treatments of MM include surgical treatment, chemotherapy, radiation therapy, immunotherapy, targeted therapy, and combination therapy, while vaccine therapy represents a new type of treatment still in clinical trials [9,10]. Surgery to remove the tumor is the primary treatment of all stages of melanoma, while chemotherapy is a cancer treatment that uses drugs to stop the growth of cancer cells, by killing the cells or by stopping their division [10]. Unfortunately, high toxicity, poor sensitivity, and debilitating side effects limit the effectiveness of these treatments [11,12,13]. Due to its very poor prognosis, chemotherapy to cure MM is often combined with other auxiliary therapies, such as biological therapy, skin-directed therapy, and radiotherapy [14]. Among the targeted therapies employed or being studied in the treatment of melanoma, signal transduction inhibitor therapy (STIT) is used to treat some patients with advanced melanoma or tumors that cannot be removed by surgery. STIT employs drugs such as BRAF inhibitors (dabrafenib, vemurafenib, encorafenib) that block the activity of proteins made by mutant BRAF genes; and MEK inhibitors (trametinib, cobimetinib, binimetinib) that block proteins (MEK1 and MEK2) which affect the growth and survival of cancer cells [10]. Combinations of BRAF and MEK inhibitors, such as dabrafenib plus trametinib, vemurafenib plus cobimetinib, and encorafenib plus binimetinib, are employed to improve the effects and reduce the dosage of the single drugs [10]. Notably, Vemurafenib (PLX4032) has been approved as the main treatment for patients with BRAF^V600E^-mutated MM [15,16]. However, despite the initial positive response, the acquisition of chemoresistance hampers the clinical efficacy of Vemurafenib. In this scenario, it is important to investigate and develop novel and efficient treatments able to counteract MM progression before metastatic spreading, especially in PLX-resistant melanoma cells, which is the scope of our current research, of which this work represents the first step. In this context, topical therapy has great potential for skin cancer due to its capacity to deliver high concentrations of bioactive compounds on tumor sites [17]. Among different formulations suitable for topical administration of drugs, composite hydrogels are acknowledged as archetypal and model platforms, due to their multifunctional properties and their capability of a sustained drug release [18,19]. Particularly, hydrogels consist of three-dimensional (3D) network structures, which can be prepared using synthetic and/or natural polymers capable of absorbing and keeping great amounts of water and/or biological fluids [20]. Additionally, biocompatible and biodegradable hydrogels could help in accelerating healing, reducing inflammation, and impeding bacterial growth, thus being suitable in the tissue engineering, pharmaceutical, and biomedical fields [19,20]. An observed disadvantage of using drug-loaded nanocomposite hydrogels consists in the risk of drug escaping and accumulation, but successful attempts have been reported to address such issues. In this regard, by modifying the hydrogels, better mechanical properties have been achieved, capable of maintaining the structural integrity of the gel and achieving the kinetic properties required for a sustained drug release [21]. In this regard, we recently reported on the hydrogel formulation of two antibacterial pyrazoles, namely, **3c** and **4b** (Appendix A), using a polystyrene-based cationic resin (R1) (Appendix A) as gelling agent [22]. R1 was instead synthesized by a one-step, low-cost, and scalable reverse-phase suspension copolymerization technique [22].

Without using additives, hydrogels were obtained in the form of micro-spherular beats, with a high equilibrium degree of swelling (EDS) and equilibrium water content (EWC). These hydrogels were demonstrated to be structurally and chemically stable over time, and to have an elastic, shear-thinning Bingham pseudoplastic behavior, as well as low yield stress values, assuring easy spreadability [22].

On the other hand, another resin (R4) whose structure is available in Appendix A was previously synthesized using the same method employed to synthesize R1 but starting from a different monomer (M4). R4 demonstrated excellent swelling properties, the capability to form a hydrogel upon simple dispersion in water, and potent bactericidal effects by contact against both Gram-positive and Gram-negative multidrug resistant (MDR) bacteria [23]. Additionally, we recently reported the synthesis, characterization, and biological evaluation of a library of imidazo-pyrazoles (IMPs), among which the compounds **4G** and **4I** (Figure 1) demonstrated interesting antiproliferative effects against several types of tumors [24].

Thus, aiming at addressing the need for novel topical formulations to treat melanoma progression by dermal administration, the cytotoxic effect of **4G** and **4I** was first evaluated on patient-isolated MM cells sensitive to PLX4032 (MEOV NT) and on in vitro selected PLX4032-resistant (MEOV PLX-R) ones [25]. The cytotoxic effects of **4G** and **4I** were investigated also on MEOV PLX-R cells because our main scope consisted of developing an imidazo-pyrazole-loaded hydrogel topical formulation active particularly on these cells, where current treatments fail. Promisingly, **4I** was significantly more effective than PLX on melanoma cells which have developed resistance upon prolonged treatments with such drug and was retained as interesting and worthy of formulation studies. Here, as a first step toward our objective above reported, **4I** was formulated as hydrogel (R4HG-4I), using the antibacterial R4 as gelling agent. The capability of R4 to self-form hydrogels allowed to obtain **4I**-loaded hydrogels without using other additives, which could alter the biological properties of **4I**, and which could be an irritant in a future dermal application. For comparison purposes, an R4-based hydrogel (R4HG) not containing **4I** was prepared using only R4 and water and was subjected to all characterization experiments carried out on R4HG-4I. Interestingly, the demonstrated antibacterial properties of R4 [23] would prevent skin infections sustained by opportunistic bacteria and favored by a relatively weak immune system of the tumor-affected skin tissue, thus aggravating melanoma intricacy, causing ulcerations, and worsening the patient’s prognosis [8]. Collectively, in this work, we have developed a hydrogel topical formulation with physicochemical characteristics supporting its suitability for dermal administration. It encompasses an antibacterial resin for limiting the emergence of infections in the melanoma lesions and the imidazo-pyrazole **4I** which was 1.4-fold more active than PLX on MEOV PLX-R cells. The topical formulation herein proposed will be essayed on a more realistic 3D model of melanoma. Upon satisfactory results, R4HG-4I could represent a platform to develop efficacious alternative strategies to treat PLX-resistant melanoma by limiting cancer cell growth and counteracting bacterial infections, also usable as a topical post-surgery device.

## 2. Materials and Methods

### 2.1. Chemicals and Instruments

All reagents and solvents were from Merck (Merk Life Science S.r.l., Milan, Italy), and they were purified by standard procedures. Ethylene bis-acrylamide (EBA) was prepared by known procedures [26], while M4 was prepared as previously described [27]. Imidazo-pyrazoles **4G** and **4I** were prepared as reported by Brullo et al. [24]. Organic solutions were dried over anhydrous magnesium sulfate and evaporated using a rotatory evaporator operating at a reduced pressure of about 10–20 mmHg. The melting ranges of solid compounds prepared in this study were determined on a 360 D melting point device, with resolution 0.1 °C (MICROTECH S.R.L., Pozzuoli, Naples, Italy). The instruments used to characterize **4G**, **4I**, and R4 used in this study and all the intermediates synthesized to prepare them were the same as previously reported [28]. The procedures and materials to perform column chromatography and thin-layer chromatography were those previously described in Alfei et al. [28]. The optical microscopy analyses were performed using a Nikon TMS-F inverted phase-contrast microscope equipped with 4× brightfield objective, 10× phase contrast objective, LWD 20× phase contrast objective, and LWD 40× phase contrast objective (Nikon Instruments, Inc., New York, NY, USA). Sequential images were acquired with 10×, 20×, and 40× phase-contrast objectives. Videos were captured with 4× and 10× phase-contrast objectives. The camera for image and video capture was a Moticam 10+ (10 MP) (MoticEurope S.L.U., Cabrera de Mar, Barcelona, Spain). Sieving was performed with a 2000 Basic Analytical Sieve Shaker-Retsch apparatus (Retsch Italia, Verder Scientific S.r.l., Pedrengo (BG), Italy). Lyophilization was performed as previously described [28], while centrifugations were performed on an ALC PK110 centrifuge, equipped with an F-G1 rotor (13.1 cm), allowing a max speed of 5000 rpm, corresponding to a relative centrifugal force (RCF) of 3661× *g* (ALC, Winchester (Hampshire), UK). Titrations were performed using an HI5522-02 Laboratory Benchtop Instrument pH/mV/ISE and EC/TDS/Salinity/Resistivity Research Grade (HANNA Instruments, Padova, Italy).

### 2.2. Chemistry

#### 2.2.1. Preparation of R4HG and R4-HG-4I

An empty R4-based hydrogel (R4HG) was first prepared inserting R4 (0.69 mL (Vi_R4HG_) 203.1 mg (Wi_R4HG_)) in a graduated centrifuge tube (Ø_est._ = 14 mm, V = 10 mL) and adding up to 10 mL of deionized water. Secondly, 0.69 mL (203.1 mg) of R4 and 0.1 mL (21.4 mg) of **4I** were inserted in an analogous tube, to obtain a mixture **4I**/R4 with initial volume Vi_R4HGI_ (0.79 mL) and initial weight Wi_R4HGI_ (224.5 mg). Then, at room temperature and under magnetic stirring, deionized water and methanol (MeOH) (qb to dissolve **4I**) were added up to a total volume of 10 mL. When homogeneous mixtures were observed, the dispersions were sonicated at 37 °C for 30 min and then degassed for 10 min, using an Ultrasonic Cleaner 220 V (VWR, Milan, Italy). The dispersions were then centrifugated at relative centrifugal force (RFC) of 2343× *g* for 30 min to separate R4HG and R4HG-4I hydrogels from water in excess, which was removed. The tubes were then turned upside down on filter paper to remove residual water and left for 10 min. The final volumes of the obtained gels (*Vf*) corresponded to the volumes of gels at their EDS. The volume of water in which **4I** + R4 or R4 alone resulted finally dispersed was determined, and the concentration (mg/mL) of R4 in R4HG, as well as the concentration (mg/mL) of **4I** and of the mixture **4I** + R4, in R4HG-4I was calculated. The initial volume (*Vi*) and the final volume (*Vf*) were used to determine the *EDS* (%) and *EWC* (%) for both gels by Equations (1) and (2).
(1)$$EDS\;\left(\%\right)=\frac{Vf-Vi}{Vi}\times 100$$
(2)$$EWC\;\left(\%\right)=\frac{Vf-Vi}{Vf}\times 100$$

R4HG and R4HG-4I were then left in the tubes carefully sealed to prevent water evaporation and stored in the fridge for subsequent characterization experiments including ATR-FTIR, water loss, potentiometric titrations, and rheological studies. A fraction of the prepared gels was lyophilized to complete their characterization by scanning electron microscopy (SEM) and to determine their swelling index.

#### 2.2.2. Scanning Electron Microscopy (SEM)

The microstructure of the lyophilized hydrogels R4HG and R4HG-4I was investigated by SEM analysis. The experiments were carried out as previously reported [22].

#### 2.2.3. Chemometric-Assisted ATR-FTIR Spectroscopy

FTIR spectra of **4I**, R4, R4HG, R4HG-4I as well as R4HG-D and R4HG-4I-D were recorded as recently described [22]. Subsequently, we first arranged the FTIR data of the spectra acquired for all the samples in a matrix 3401 × 6 of 20,406 measurable variables, then after the removal of spectral data of swollen R4HG and R4HG-4I, we arranged the residual data in a smaller matrix 3401 × 4 of 13,604 measurable variables. The obtained matrices were processed by Principal Components Analysis (PCA), using CAT statistical software (Chemometric Agile Tool, free down-loadable online, at: http://www.gruppochemiometria.it/index.php/software/19-download-the-r-based-chemometric-software; accessed on 29 August 2023).

#### 2.2.4. Content of **4I** in R4HG-4I, Drug Loading (DL%) and Entrapment Efficiency (EE%)

First, the **4I** calibration curve was constructed. A stock solution of **4I** (2 mg/mL; 4.3 mM) was prepared dissolving, **4I** (10 mg) in MeOH (5 mL), applying ultrasounds and heating. The obtained solution was added with 1 mL of Schiff′ fuchsin-sulfite reagent (Merck Italy, Milan, Italy) and stirred at room temperature to reach a magenta coloration, as indicated in the literature [29]. Then, upon proper dilutions with MeOH, standard solutions at **4I** concentrations of 0.3, 0.25, 0.08, and 0.06 mg/mL were obtained. The **4I** solutions were analyzed using a Fiber Optic UV-Vis Spectrometer System Ocean Optics USB 2000 (Ocean Optics, Inc., Dunedin, FL, USA) in 3 mL quartz cuvettes, thus detecting the related absorbance (Abs) at room temperature and at ʎ abs = 543.88 nm. A solution of MeOH and Schiff reagent not containing **4I** was used as blank. Determinations were made in triplicate, and results were reported as mean of three independent experiments ± SD. The **4I** concentrations were plotted vs. the Abs values, and the **4I** calibration curve (Appendix A) was obtained by least-squares linear regression analysis using Microsoft Excel 365 software. Equation (3) of the developed linear calibration model was the following.
(3)y=0.503x+0.599

In Equation (3), *y* is the Abs values measured at ʎ abs = 543.88 nm and *x* the **4I** standard concentrations analyzed. In Equation (3), the slope represents the coefficient of extinction (ε) of **4I**.

To estimate the **4I** contained in R4HG-4I, 2 g of swollen hydrogel were lyophilized obtaining 82.6 mg of fully dried gel, which were dispersed in 10 mL of MeOH and vigorously stirred for ten minutes to promote the release of **4I**. Aliquots of the clear solution were treated with the Schiff′ fuchsin-sulfite reagent (Merck Italy, Milan, Italy) as previously reported, and the amount of **4I** in the aliquots was quantified at ʎ abs = 543.88 nm by UV–vis analysis as reported above. Determinations were made in triplicate, and results were reported as mean of three independent experiments ± SD.

The values of *DL%* and *EE%* of R4HG-4I were calculated from the following Equations (4) and (5) [30,31].
(4)$$DL\;\left(\%\right)=\frac{Weight\;of\;4I\;in\;the\;Gel}{Weight\;of\;Gel}\times 100$$
(5)$$EE\;or\;DLC\;\left(\%\right)=\frac{Weight\;of\;4I\;in\;the\;Gel}{Weight\;of\;4I}\times 100$$

##### Statistical Analysis

The statistical significance of the slope of the **4I**-calibration curve was investigated through the analysis of variance (ANOVA), performing the Fischer test. Statistical significance was established at the *p*-value < 0.05.

#### 2.2.5. Biodegradability of R4HG and R4HG-4I over Time by In Vitro Mass Loss Experiments

Samples of vacuum-dried hydrogels were weighted, inserted in centrifuge tubes, added with 10 mL of PBS, and incubated at 37 °C. At fixed time points of 4, 8, 14, and 21 h, then 1, 2, 3, 4, 6, and 8 days, hydrogels in the tubes were centrifugated to remove the supernatant PBS, and the water on the hydrogel surface was wiped off, by inverting the tubes.

The hydrogels were then weighed to record their mass change with time. The weight of the hydrogel at the preset time intervals has been indicated as *Mt*. The cumulative mass loss (*ML*) percentages (mass change) over time were calculated according to the following Equation (6):(6)$$Mass\;Change\left(ML\right)\left(\%\right)=\frac{Mi-Mt}{Mi}\times 100$$
where *Mi* and *Mt* are the initial mass of the swollen resin and its mass after a time *t* in PBS, respectively.

#### 2.2.6. Water Loss Tests

Samples of the freshly prepared R4HG and R4HG-4I exactly weighted (556.9 mg and 504.3 mg, respectively) were deposited in watch glasses, which were then placed in an oven at 37 °C. The water loss was checked over time until the weight was constant. The cumulative water loss percentages were determined by Equation (7):(7)$$Water\;Loss\;\left(\%\right)=\frac{Wi-Wt}{Wi}\times 100$$
where *Wi* and *Wt* are the initial weight of the swollen resin and its weight after a time *t*, respectively.

#### 2.2.7. Equilibrium Swelling Rate

The swelling measurements were carried out as previously described using 42.2 mg of fully dried R4HG (R4HG-D) and 47.4 mg of fully dried R4HG-4I (R4HG-4I-D) [22]. In this case, measurements were made at intervals of time of 15 min until the weight was constant. The cumulative swelling ratio percentage (*Q* %) as function of time was calculated from Equation (8).
(8)$$Q\;\left(\%\right)=\frac{WSt-WD}{WD}\times 100$$
where *WD* and *WSt* are the weights of the lyophilized gel and those of the swollen gel at time *t*, respectively. The equilibrium swelling ratio (*Q*_equil_) was determined at the point (time *t*) the hydrated gels achieved a constant weight.

#### 2.2.8. Potentiometric Titration of R4HG and R4HG-4I

Potentiometric titrations were performed on R4HG and R4HG-4b as previously described [22] using 50.3 mg of R4HG and 52.8 mg of R4HG-4I dispersed in 50 mL of Milli-Q water (m-Q). Briefly, they were magnetically stirred and added with a standard 0.1 N NaOH aqueous solution (2.0 mL, pH = 10.34 for R4HG and pH = 11.20 for R4HG-4I), and potentiometrically titrated by the addition of aliquots (0.2 mL) of HCl 0.1 N to reach a volume of 3.0 mL [28].

Titrations were performed in triplicate, and the obtained pH values were expressed as mean ± SD.

#### 2.2.9. Rheological Experiments

The rheological properties of R4HG and R4HG-4I were assayed according to procedures previously reported [22].

#### 2.2.10. Evaluation of **4I** in Vitro Releases

The in vitro release of **4I** from R4HG-4I was determined by weighing about 1 g of hydrogel which was lyophilized obtaining 41.3 mg of fully dried R4HG-4I (R4HG-4I-D) whose determined drug loading was 8.4%. The sample was immersed in 10 mL of PBS medium (pH = 7.4) in a dialysis bag with 3.5 K MWCO (Cellu Sep H1, Orange Scientific, Braine-l’Alleud, Belgium) and subsequently dialyzed against 30 mL of release medium (PBS) at 37 °C and 100 rpm. At fixed interval points (0, 1, 2, 4, 24, 28, 48, and 72 h), aliquots of 2 mL of the release medium were taken out and 2 mL of fresh PBS were replenished. Each aliquot was treated with the Schiff′ fuchsin-sulfite reagent (Merck Italy, Milan, Italy) as previously described. The amount of **4I** in the aliquots was detected by measuring their absorbance at 543.88 nm using the UV–vis spectrophotometer previously described, while a PBS solution not containing **4I** and treated with the Schiff reagent was used as blank. Determinations were made in triplicate, and results were expressed as mean of three independent experiments ± SD. The obtained concentrations were used to compute the cumulative drug release percentage, according to Equation (9):(9)$$CDR\;\left(\%\right)=\frac{Dt}{Di}\times 100$$

In Equation (9), *Dt* is the amount of **4I** released at incubation time *t*, while *Di* is the **4I** concentration in the dialysis tube, given by the total **4I** entrapped in the weight of R4HG-4I analyzed according to the computed DL% when dissolved in 10 mL of PBS. In parallel, the **4I** release from a **4I** suspension in PBS at **4I** concentration as in the previous experiment was determined in the same condition and following the same procedure above described.

### 2.3. Biological Screening

#### 2.3.1. Cell Culture Conditions

MEOV NT cell line, directly obtained from the biopsy of an untreated patient with metastatic melanoma (MM), was kindly provided by Prof. Gabriella Pietra (University of Genoa, Genoa, Italy). The PLX4032-resistant cell line (MEOV PLX-R) was selected by treating MEOV NT cells for 6 months with increasing concentrations of PLX4032 [25]. Briefly, the drug-resistant population was obtained by seeding the MEOV NT cells twice a week at a density of 1.5 × 10^6^ and treated over six months with PLX4032 (250 nM–1.5 µM). The authenticity of the selected cells was checked by Short Tandem Repeat (STR) profile analysis performed by the Immunohematology and Transfusion operative unit, IRCCS Ospedale Policlinico San Martino, Genoa. Both cell lines were maintained in RPMI 1640 medium (Euroclone Spa, Pavia, Italy) supplemented with 10% Fetal Bovine Serum (FBS, Euroclone Spa, Pavia, Italy), 1% L-Glutamine (Euroclone Spa, Pavia, Italy), and 1% Penicillin/Streptomicin (Euroclone Spa, Pavia, Italy) and grown in standard conditions (37 °C humidified incubator with 5% CO_2_).

#### 2.3.2. Treatments

MEOV NT and MEOV PLX-R were treated for 24, 48, and 72 h with increasing concentrations (0–100 µM) of **4G** and **4I**. Cell cultures were carefully monitored before and during the experiments to ensure optimal cell density. Notably, samples were discarded if the cell confluence reached >90%.

#### 2.3.3. Cell Viability Assay

Cell viability was determined by using the CellTiter 96^®^ AQueous One Solution Cell Proliferation Assay (Promega, Madison, WI, USA). Briefly, cells (10,000 cells/well) were seeded into 96-well plates (Corning Incorporated, Corning, NY, USA) and then treated. Next, cells were incubated with 20 µL of CellTiter, and the absorbance at 490 nm was recorded using a microplate reader (EL-808, BIO-TEK Instruments Inc., Winooski, VT, USA). IC_50_ was evaluated by GraphPad Prism 5.4.2 Software (GraphPad Software, Boston, MA, USA).

#### 2.3.4. Detection of Hydrogen Peroxide (H_2_O_2_) Production

The production of H_2_O_2_ was evaluated using 2′-7′-dichlorofluorescein-diacetate (DCFH-DA; Merk Life Science S.r.l. Milan, Italy) as previously reported [32].

#### 2.3.5. Statistical Analyses

Data are expressed as means ± SEM of at least three independent experiments in which six different wells were analyzed every time for each experimental condition. In the analyses of cell viability or H_2_O_2_ levels, the condition of untreated cells was set as 100% ± SEM. Statistical significance of differences was determined by one-way or two-way analysis of variances (ANOVA); *p* < 0.05 was considered statistically significant. When the two-way ANOVA was performed, the Bonferroni post hoc tests were carried out, using GraphPad Prism 5 (GraphPad Software v5.0, San Diego, CA, USA). Asterisks or other indicators (see Figures captions) indicate the following *p*-value ranges: * = *p* < 0.05, ** = *p* < 0.01, *** = *p* < 0.001.

## 3. Results and Discussion

### 3.1. Chemistry

#### 3.1.1. Synthesis and Characterization of R4

Resin R4 has been prepared as previously reported, by reverse suspension polymerization at 35 °C [23], according to Figure 1.

During the polymerization reaction, the precipitated R4 was recovered by filtration, and the filtrate was washed several times using proper solvents as previously described [23]. R4 was left at reduced pressure until a constant weight was reached (conversion yield of 98%). Following the reported protocol, R4 was fractioned by sieving using sieves with 25–120 mesh, obtaining a grainy material (about 96% of the original weight). R4 here prepared was characterized by ATR-FTIR analyses, while the equivalents of NH_3_^+^ contained in the resin were determined by potentiometric titration. The ATR-FTIR spectra and results from the determination of NH_3_^+^ groups were like those reported previously [23], while in the optical images of the sieved-dried R4, particles appeared as micro-spherular beads in the size range 125–500 µm. Appendix A collects the results related to the content of the NH_3_^+^ group in R4, which were like those reported previously [23].

#### 3.1.2. Preparation of R4HG and R4HG-4I

According to a previous report, R4 possesses the high capability to soak up water and the capacity to give hydrogels (R4HG) with an equilibrium degree of swelling (EDS) of 731% and equilibrium water content (EWC) of 88% [23]. Additionally, the fully dried gel obtained by water loss experiments was capable of reaching very rapidly (15 min) a very high value of maximum swelling capability (3595%) [23]. Finally, in time–kill experiments on clinical isolates of multidrug-resistant (MDR) pathogens, R4 showed rapid bactericidal or bacteriostatic effects depending on the bacterial species [23]. In this regard, if used as an ingredient of topical formulations designed to treat MM, due to its established antibacterial properties, R4 may prevent possible skin infections by opportunistic bacteria, which may cause ulcerations, thus aggravating further the melanoma intricacy and improving the mortality of patients [8].

Based on these data, following the procedure recently described [22,33], we firstly prepared an empty R4-based hydrogel (R4HG). Secondly, aiming at developing a novel topical formulation to treat MM and prevent complications by infections, we considered the imidazo-pyrazoles **4I** and **4G** shown in Figure 1, previously found active against several cancer cell lines [24], as possible bioactive ingredients to prepare an imidazo-pyrazole-loaded hydrogel using the antibacterial R4 as gelling agent. Compound **4I** was preferred to its analogue **4G,** based on the results reported in the biological section of this study. Particularly, it resulted more active than **4G** in reducing the viability both in PLX-sensitive (MEOV NT) and PLX-resistant (MEOV PLX-R) MM cells, as reported in Table 1. Additionally, **4I** resulted significantly (two-sample Student *t* test) more effective than PLX, the main drug currently available to counteract melanoma, in melanoma cells which have developed resistance upon continued treatment with such drug.

Briefly, we prepared R4HG and R4HG-4I hydrogels, achieving 3D networks containing the maximum amount of water that either R4 or the mixture R4/**4I** were capable of soaking up. Table 2 collects the experimental data of the preparation of R4HG from R4 and of R4HG-4I from **4I** + R4, their EDS and EWC, as well as the details about the ingredients’ concentrations in the prepared hydrogels.

As reported in Table 2, the concentration of R4 in R4HG was 53.3 mg/mL (5.3% *wt*/*v*), while those of **4I** and R4 + **4I** in R4HG-4I were 4.1 mg/mL (0.4% *wt*/*v*, 8.8 mM) (**4I**) and 43.1 mg/mL (4.3% *wt*/*v*) (R4 + **4I**), respectively. To forecast a possible activity of R4HG-4I, we consider the calculated concentration of **4I** in the obtained R4HG-4I hydrogel and the results obtained in dose- and time-dependent cytotoxicity experiments on MM cells with not-formulated **4I**. We found that concentrations of **4I** 1000 times lower than those calculated for the obtained gel caused a significant reduction in MEOV NT at 24, 48, and 72 h of exposition, while in MEOV PLX-R cells at 48 and 72 h of exposure. Specifically, the viable MEOV NT cells were 79%, 61%, and 49%, at 24, 48, and 72 h, respectively, while the viable MEOV PLX-R cells were 68 and 54% at 48 and 72 h, respectively when **4I** was administered at a concentration of 8.7 µM. In addition, in experiments carried out to assess the production of ROS in MM cells treated with **4I**, **4I** concentrations in the range 8–10 µM caused a significant increase of ROS in both cell populations. Regardless of the calculated concentration of **4I** reported in Table 2, the exact amount of **4I** in R4HG-4I gel was also determined by UV–vis analysis.

In the Appendix A shows the obtained hydrogels R4HG (Appendix A) and R4HG-4I (Appendix A). The observable volume of hydrogels (Appendix A), also reported in Table 2, was consistent with that of gels at their EDS (%). In Appendix A, the hydrogels can be observed both in vertical and inverted positions. The following Figure 2a–d show representative images of R4HG after sonication and drying at 100 °C until a constant weight equal to that before swelling was achieved (R4HG-D), captured with an optical microscope. Particularly, the optical micrographs obtained with objective from 10× to 40× evidenced aggregates of pseudo-spherical particles with sizes in the range 200–400 µm.

Figure 3a–d show the fully dried hydrogel R4HG-4I-D obtained after drying R4HG-4I at 100 °C until a constant weight equal to that before swelling was achieved.

In the optical micrographs obtained with an objective from 10× to 40×, R4HG-4I showed aggregates of pseudo-spherical particles. In this case, measurable spherical particles showed smaller sizes in the range 170–265 µm. In the Appendix A, further optical images of R4HG-D and R4HG-4I-D are available (Appendix A).

The EWC (%) of R4HG (85%) and R4HG-4I (87%), indicating the hydrogel porosity [22,23], as well as their EDS (%) by volume (552% and 675%, respectively) were very high. Interestingly, their EDS% or swelling ratio, expressing their capability of soaking up water or, possibly, wound exudate, may endow to the hydrogels with high potentiality to function as wound-healing agents [34]. In fact, the EDS% determined for our hydrogels was much higher than those recently reported by Cheng et al. for their GelMA-DOPA hydrogels (8.3–20.2%), which were successful anyway in promoting significant wound repair and cell proliferation [35]. In the Appendix A, we have provided links to videos showing the rapid process of swelling and the high absorption capability of dried R4HG-D (objective 10×) and R4HG-4I-D (objective 4×), after the addition of a drop of water. Particularly, Appendix A (https://clipchamp.com/watch/LVAKVdP1F8f), (accessed on 26 September 2023), regarding the addition of water to the aggregate of R4HG-D showed in Figure 2a,d, evidenced as, upon contact with water, the particles of dried gel rapidly absorbed water with evident swelling. Initially, swelling highlighted more defined small spherical particles in motion which, due to further water absorption, become a shapeless highly swollen brilliant globular mass completely soaked in water (Appendix A). As shown in Appendix A, regarding the 4I-enriched R4HG dried gel, its swelling rate was even higher than that of RGHG-D, as later confirmed by the equilibrium swelling rate experiments. Particularly, in Appendix A (https://clipchamp.com/watch/TtDuSbTIuWF), (accessed on 26 September 2023), it is observable as the drop of water coming from the right side, upon contact with the aggregates of the dried gel was rapidly absorbed by them which underwent a so-fast swelling, to make it impossible to detect spherical particles in motion, as in the case of R4HG-D.

#### 3.1.3. SEM Analysis

The morphology of lyophilized R4HG and R4HG-4I was investigated also by SEM. Figure 4 shows the microstructure of R4HG (Figure 4a) and R4HG-4I (Figure 4b).

The SEM micrograph of R4HG did not evidence spherical beads, rather, an irregular morphology characterized by large inner cavities, like that of the doxorubicin (DOX)-loaded hydrogel reported by Li et al. [36]. On the contrary, R4HG-4I showed spherical beads of about 150–200 µm attached to one each other and significantly narrower cavities.

#### 3.1.4. Chemometric-Assisted ATR-FTIR Analyses

ATR-FTIR analyses were carried out on the compounds **4I** and R4, used as ingredients of the hydrogel formulations, on the swollen hydrogels R4HG, R4HG-4I, as well as on the fully dried gels R4HG-D and R4-HG-4I-D, to have qualitative indication of the composition of the developed formulations. In the Appendix A show a comparison between the spectra of ingredients (R4 in Appendix A, and both R4 and **4I** in Appendix A) and the related fully dried gels (R4HG-D in Appendix A and R4HG-4I-D in Appendix A). Appendix A shows instead a comparison between the empty swollen hydrogel R4HG and the **4I**-loaded swollen hydrogel R4-HG-4I.

Particularly, Appendix A highlights as the spectrum of the fully dried gel R4HG-D was identical to that of its unique ingredient R4, showing a large band at 3200–3600 cm^−1^ (NH stretching) and bands at 2927/2029 cm^−1^ (alkyl CH stretching), 1611 cm^−1^ (amide C=O stretching) 1498 cm^−1^ (C=C stretching of the aromatic ring), and 1056 cm^−1^ (in plane CH banding). This result established that during swelling in water, no alteration in the functional groups of R4 has occurred and that upon drying the original R4 can be recovered, confirming the total reversibility of the swelling process. On the other hand, Appendix A evidenced that also the spectrum of the fully dried gel which should contain **4I** is practically identical to that of R4 (and to that of the empty R4HG-D), and only very small bands indicating the presence of **4I** were detectable in the region of spectrum between 1000 and 600 cm^−1^. On the contrary, the spectra of swollen R4HG and R4HG-4I containing high percentages of water evidenced mainly the two bands peculiar to H_2_O molecules. A very large and intense band at 3700–2900 cm^−1^ (OH stretching) and another band at 1635/1636 cm^−1^ (OH bending) constituted the very simple and similar spectra of the two hydrogels. Anyway, in the spectrum of R4HG-4I, a band at 1016 cm^−1^ indicating the presence of **4I** was observable.

##### Principal Components Analysis (PCA)

Since just by observing the ATR-FTIR spectra in Appendix A the presence of **4I** in R4HG-4I-D gel was not clearly confirmed, to have more reliable information from spectral data, we processed the ATR-FTIR spectral data of all samples using the principal components analysis (PCA) [37]. PCA allowed to envisage the location taken up by **4I** R4, R4HG, R4HG-4I, R4HG-D, and R4-HG-4I-D in the scores plot of PC2 (explaining the 36.3% of variance) vs. PC1 (explaining the 59.7% of variance) (Appendix A), knowing that samples located close to each other along either component PC1 or PC2 are structurally similar, while those placed far apart are structurally different. As observable, PC1 discriminated between polymeric compounds containing resin R4, located at positive scores, and the small molecule **4I** located at negative scores, while on PC2 samples, they were clustered on the base of their water content. Particularly, at positive scores were located the swollen gels with high content of water, while at negative scores we found the dried materials not containing water. Among the swollen gels R4HG and R4HG-4I, the latter was positioned slightly closer to **4I** than R4HG, confirming the presence of small quantities of **4I** in R4HG-4I. Similarly, among dried polymeric compounds, R4HG-D was positioned closer to R4 than R4HG-4I-D, confirming the presence of higher quantities of R4 in R4HG-D than in R4HG-4I-D. For further confirmation of the presence of **4I** in R4HG-4I-D, we simplified the data set removing the spectral data of the swollen hydrogels and reperformed PCA on the residual data. Appendix A shows the obtained new score plot of PC1 vs. PC2.

As in the previous experiment, PC1 discriminated between polymeric samples located at positive scores and the imidazo-pyrazole positioned at negative scores. Interestingly, this time on PC2 we can observe that both **4I** and R4HG-4I-D were positioned more distant than R4HG-D from R4, thus indicating structural similarities between R4 and R4HG-D and between **4I** and R4HG-4I-D and confirming the presence of **4I** in the latter.

#### 3.1.5. Determination of the Exact Amount of **4I** Loaded in R4HG-4I

The amount of **4I** contained in the fully dried hydrogel was measured by means of a UV-vis spectrophotometer, against a previously prepared calibration curve. In this regard, since in early experiments analyzing colorless **4I** solutions we found that no absorbance peck was detectable in the wavelength range 200–900 nm, we reacted the **4I** stock solution prepared for developing the calibration model with the Schiff′ fuchsin-sulfite reagent. We obtained a magenta solution, and following the procedure described in the experimental section, the **4I** calibration model shown in Appendix A was developed by ordinary least squares (OLS) method. Additionally, Appendix A shows a representative UV–vis spectrum of a **4I** methanol solution upon treatment with the Schiff′ reagent.

Although the high value of the coefficient of determination (R^2^ = 0.9944) assured the linearity of calibration, we further assessed the linearity and sensitivity of the developed calibration model, verifying the statistical significance of its slope. To this end, we performed the analysis of variance (ANOVA) by the Fischer test, and the statistical significance was established at the *p*-value < 0.05. The developed calibration model was used to determine the amount of **4I** contained in R4HG-4I, as well as the values of drug loading (DL%) and entrapment efficiency (EE%), also called drug-loading capacity (DLC%), as described in the Experimental Section. Table 3 reports the obtained Abs and the computed **4I** concentrations.

Upon proper calculations, it was established that the content of **4I** in the total amount of R4HG-4I prepared (5.45 g when swollen and 0.2245 when fully dried) was 18.9 mg, DL (%) was 8.4%, and the EE (%) was 88.4%. In this regard, the EE% of R4HG-4I was higher than the optimized EE% recently obtained by Soni and Yadav (86%) for poloxamer-based injectable thermos-responsive hydrogels loaded with etoposide [38]. Furthermore, although we did not use any drug-binding effector as graphene oxide (GO), the EE% of R4HG-4I was also better than that of both fluorouracil-loaded pH-sensitive konjac glucomannan/sodium alginate (KGM/SA) and KGM/SA/graphene oxide (KGM/SA/GO) (10.29%, 13.32%, 15.50%, 17.49%, 22.73%, 27.09%, and 32.04%) reported by Wang et al. [39]. Additionally, the EE% or DLC (%) of R4HG-4I was higher than that of the nine β-cyclodextrin/polyvinypyrrolidone-co-poly (2-acrylamide-2-methylpropane sulphonic acid) hybrid nanogels formulation prepared by Shoukat et al., which demonstrated DLC% in the range of 67–89% [30].

Due to the presence of several hetero atoms such as nitrogen and oxygen atoms in the structure of R4 and **4I**, we assume that the drug loading occurred mainly based on hydrogen bonding and surface adsorption.

#### 3.1.6. Evaluation of Biodegradability of R4HG and R4-HG-4I over Time by Mass Loss Experiments

Biodegradability is an essential property that hydrogels finalized for biomedical applications should possess [40]. So, to evaluate the biodegradability over time in PBS solution of both R4HG and R4HG-4I, we measured their mass loss with time by recording their weight change after a specified time of incubation in PBS, as reported in the literature [41] and described in the Experimental Section. The cumulative mass loss percentages over time were determined according to the equation reported in the Experimental Section and plotted against times, achieving the lines in Figure 5a. All experiments were performed in triplicate, and results were expressed as mean ± standard deviation (SD).

As observable in Figure 5a, the mass loss profile was similar for both empty R4HG and 4I-enriched R4HG-4I, but the total mass loss of R4HG-4I was slightly higher than that of R4HG (96% vs. 84%), thus establishing that R4HG-4I is more biodegradable than R4HG. The equilibrium was reached by both hydrogels after 3 days, but initial degradation was more rapid for R4HG-4I than for R4HG, and the 82% mass loss was observed after just a day. In this regard, the degradation of our hydrogels was more rapid and quantitative than that of all PEG/lecithin–liquid–crystalline composite hydrogels developed by Zhang et al., which reached the equilibrium after about 8 days with a mass loss in the range of 25–55% [41]. The mass loss profiles of R4HG and R4HG-4I were further studied by fitting the data of the curves in Figure 5a with the most common mathematical kinetic models including the zero-order model, first-order model, Hixson–Crowell model, Higuchi model, and Korsmeyer–Peppas model. The related dispersion graphs were obtained, which were helpful to precisely establish the kinetics and the main mechanisms that govern the degradation over time of our hydrogels [42,43]. Table 4 reports the coefficients of determination (R^2^) of the equations of the linear regressions of the obtained dispersion graphs, considered as the parameters to determine which model best fits the mass loss data according to the literature [42]. Since the Korsmeyer–Peppas model provided the highest values of R^2^, it was established that the degradation of both hydrogels best fit with the Korsmeyer–Peppas kinetic model (Figure 5b).

Specifically, Korsmeyer–Peppas kinetics describes releases (drugs, water, mass) from polymeric systems by Equation (10) [44]:(10)$$\ln\left(Mt\right)=n\times \ln\left(t\right)+\ln\left(K\right)$$
where *Mt* is the amount of mass lost from the hydrogel on time *t*, (*ln*(*Mt*) corresponds to the variable *y* in the equation in Figure 5b, *Ln*(*t*) corresponds to the variable *x* in the equation in Figure 5b, *K* is the transport constant which incorporates structural/geometrical information of the polymer system, while *n* (also called diffusional or transport exponent) provides information on the possible mechanism(s) of the mass loss and degradation (Fickian diffusion or non-Fickian diffusion) [45]. Accordingly, in our case, *n* corresponded to the slopes of the linear regressions of the Korsmeyer–Peppas mathematical models shown in Figure 5b, while *Ln*(*K*) was their intercept. Consequently, *K* were in both cases positive and equal to 32.40 for R4HG and 49.92 for R4HG-4I, while the values of *n* were equal to 0.5723 (R4HG) and 0.7524 (R4HG-4I). R4HG-4I demonstrated a value of *K* higher than that of R4HG, thus asserting that the degradation of R4HG-4I is faster than that of R4HG. Additionally, since when 0.5 < *n* < 1 the model is non-Fickian but anomalous, the mechanism of mass loss from both empty R4HG and **4I**-enriched R4HG-**4I** was governed by both diffusion and swelling. Particularly, the slow rearrangement of polymeric chains occurring upon swelling and the simultaneous diffusion process caused the time-dependent anomalous effects [46].

#### 3.1.7. Water Loss Experiments

The water loss profiles of R4HG and R4HG-4I were obtained by heating them at 37 °C over time, and measuring their weight loss, until an approximately constant weight was reached (420 min), and the fully dried R4HG-D and R4HG-4I-D were achieved. Such experiments were performed to evaluate the capability of the prepared hydrogels to retain water when air-exposed and therefore a possible exudate in the melanoma ulcerations, and the time necessary for the gels to reach full dryness thus losing its spreadability.

By plotting the cumulative water loss percentages of the two hydrogels vs. times, the graphs shown in Figure 6a were obtained.

As reported Figure 6a, the water loss was similar for both empty R4HG and 4I-enriched R4HG-4I (95% and 96%, respectively). The equilibrium was reached by both hydrogels after 5 h, thus demonstrating high capability to retain water and therefore a possible exudate of the melanoma ulceration. The water loss profiles of R4HG and R4HG-4I were further studied by fitting the data of the curves in Figure 6a with the most common mathematical kinetic models as described in the previous Section 3.1.6 [42,43]. Appendix A reports the coefficients of determination (R^2^) of the equations of the linear regressions of the obtained dispersion graphs, which established that, as for the degradation processes, the water loss from both hydrogels best fit with the Korsmeyer–Peppas kinetic model described by Equation (10) (Figure 6b).

*K* values were positive and equal to 1.7998 for R4HG and 3.1068 for R4HG-4I, while the values of *n* were equal to 0.6887 (R4HG) and 0.6011 (R4HG-4I). The lower values of *K* evidenced that degradation vs. time in solution occurred faster than the release of water under heating and that the water release was faster in R4HG-4I than in R4HG. In both cases, the water loss was non-Fickian but anomalous, and the mechanism of water release from both empty R4HG and **4I**-enriched R4HG was governed by more than one release phenomenon type, whose rates are almost comparable [46].

#### 3.1.8. Equilibrium Swelling Rate Experiments

The swelling of a drug delivery system has enormous and direct influence on drug release characteristics, so it was important to investigate the swelling properties of the developed R4HG-4I and for comparison those of empty R4HG [31]. The swelling measurements were made using samples of the fully dried R4HG-D and R4HG-4I-d in deionized water, at 37 °C every 15 min until the weight of the swollen resins recovered by centrifugation was approximately constant or started to decrease.

Figure 7 shows the cumulative swelling ratio (%) graphs of R4HG and R4HG-4I.

According to a composite pyrazole-enriched polystyrene-based hydrogel recently reported by us [22], the maximum swelling of R4HG-4I (2921%) was observed in only 15 min, due to the hydrophilic nature of its ingredients and to its spongy characteristic. These high swelling capabilities, as confirmed later by the release experiments, should favor the rapid drug (**4I**) release in aqueous medium. Since the equilibrium swelling ratio (*Q*_equil_) is defined as the point at which the hydrated resin achieves a constant weight, for the **4I**-loaded R4HG it was not possible to determine an exact value of *Q*_equil_. In fact, once the maximum swelling was attained, the reached weight did not remain constant, but upon further treatments with water, a reduction in the weight of the hydrogel was observed. This decreasing in weight evidenced some degradation process, thus confirming the biodegradability of R4HG-4I established in the previous experiments (Section 3.1.6), as well as the release of **4I**. Collectively, like the higher swelling capacity, also this early degradation could allow a remarkable release of **4I**, at least over the following 30 min. After this time, the decrease in weight significantly decreased because it was sustained mainly by the biodegradation of the 3D network of R4HG-4I occurring over days. As already noted, the ability of R4HG-4I-D to absorb large quantities of water so quickly established its great potential in the treatment of skin MM, especially in the presence of infections and exudates. Concerning empty R4HG-D, it demonstrated a swelling profile very different from that observed for R4HG-4I-D, thus establishing that the presence of **4I** strongly affected the capability of dried hydrogels to absorb water. As reported [47], the composition of hydrogels strongly affects their swelling behavior. In our case, it can be assumed that the presence of **4I** in the 3D network of R4HG, having several heteroatoms capable of giving hydrogen bonds with water, improved the capability of the hydrogel to interact with water, to absorb it, and its swelling rate resulted increased. Precisely, the absorption of water by R4HG-D was significantly slower than that by R4HG-4I, as well as its previously reported biodegradation, and the maximum value of swelling capability, which was minor compared to that of R4HG-4I and equal to 1806% (vs. 2921%), was reached after 45 min. In this case, after this point, a slight reduction in the weight of swollen resin was observed for further treatment with water, evidencing initial biodegradation.

##### Kinetic Studies

To obtain valuable information on the mechanisms ruling the water adsorption and the swelling processes of our dried hydrogels, we carried out a kinetic study following the procedure recently reported for other adsorption processes [33,48]. Particularly, we applied kinetic models of pseudo-first-order (PFO), pseudo-second-order (PSO), and intra-particle diffusion (IPD) (or Higuchi) to the data of cumulative swelling (%) curves reported in Figure 7. Following, we have reported the Equations (11)–(13) describing the abovementioned models.
(11)$$Ln\left(We-Wt\right)=lnWe-K1\times t$$
(12)$$\frac{t}{Wt}=\frac{1}{K2\times We^{2}}+\frac{1}{We}t\;$$
(13)$$Wt=Kint\times t^{0.5}+I$$

In the equations, *We* and *Wt* are the weight of R4HG-D and R4HG-4I-D at the maximum adsorption and at time *t*, respectively, *K*1 is the adsorption constant of the PFO kinetic model (1/min), *K*2 is the equilibrium constant velocity of the PSO kinetic model (g/mg × g), *K int* is the IPD rate constant (mg/g × min^0.5^), and *I* is the intercept of the linear curve (mg/g). Values of *ln*(*We* − *Wt*), *t*/*Wt,* and *Wt* were plotted vs. times, times again, and the root square of times, respectively. Dispersion graphs were obtained, and their linear regression lines were provided by Microsoft Excel software 365 using the Ordinary Least Squares (OLS) method (Appendix A). The coefficients of determination (R^2^) of all the equations of the linear regressions obtained have been reported in Appendix A. As in the previous Section 3.1.6 and Section 3.1.7, R^2^ values were the parameters for determining the kinetic models that best fit the data of the swelling processes and the mechanisms that governed the water absorption reactions. As often reported [33,48], the results showed that, in both cases, the R^2^ values for the PSO models were higher than those obtained for both the PFO and IPD models. Consequently, the kinetic behavior of the water absorption and swelling of both R4HG and R4HG-4I followed the PSO model.

Particularly, Appendix A shows the PSO kinetic models of water adsorption and swelling of R4HG and R4HG-4I.

In adsorption ruled by PSO kinetics, chemical adsorption involving the sharing or exchange of electrons between the adsorbent and water or electrostatic interactions is the main mechanism by which water is adsorbed [33,48].

The values of *K2* and *We* (mathematical model) (*We*
_PSO_) were computed using the values of the slopes and intercepts of the equations in Appendix A for all experiments and included in Table 5.

The values of *K2* confirmed that R4HG-4I-D was able to adsorb water faster than R4HG-D. Additionally, while the calculated value of *We* for R4HG (3.9139) perfectly agreed with the experimental one (3.9081), thus establishing that diffusion stage within the particle was not involved in the mechanism of water adsorption by R4HG-D, *We*
_PSO_ for R4HG-4I (0.4286) was significantly lower than the experimental data (0.6778), thus establishing that, probably, a diffusion stage within the particle was involved in the mechanism of water adsorption by R4HG-4I-D [49].

#### 3.1.9. Potentiometric Titrations of R4HG and R4HG-4I

Potentiometric titrations of R4HG and R4HG-4I were performed to determine the absolute amount of cationic ammonium groups deriving by R4 present in the prepared hydrogels. These data are important for the antibacterial properties of cationic materials [23] and to forecast those of the prepared hydrogels. Additionally, a high number of cationic groups could improve the cytotoxicity of **4I** by allowing electrostatic interactions with the negatively charged surface of tumor cells, thus damaging cells membrane and triggering cells death [50]. The measurements of pH upon the addition of aliquots of HCl 0.1N (0.2 mL) were carried out as described in the Experimental Section. Here, by plotting the measured pH values vs. the aliquots of HCl 0.1N added, the titration curves of both R4HG and R4HG-4I were obtained (Figure 8, continuous lines with round indicators). Subsequently, we obtained their first-derivative (FD) curves by computing the values of dpH/dV values and plotting them in the same graph vs. the corresponding volumes of HCl 0.1 N (Figure 8, dotted lines with square indicators). The maxima of the FD lines indicate different phases of the protonation process and correspond to different titration endpoints.

The titration profile (error bars omitted to not complicate the image) was similar for R4HG and R4HG-4I. The representative titration curves that are shown in Figure 8 evidenced that the potentiometric titration of both R4HG (red line) and R4HG-4I (blue line) exhibited two endpoints, the first one upon the adding of 0.6 mL of HCl 0.1N for both samples (maxima 12 (R4HG) and 12.75 (R4HG-4I)), and the second one upon the addition of 1 mL HCl 0.1N for both samples as well (maxima 7.5 (R4HG) and 8.75 (R4HG-4I)). As reported by the group of Zu in 2021 [51], while the first endpoints with higher maxima corresponded to the titration of any excess NaOH, the second ones can be ascribed to the protonation of the amine groups. Accordingly, the number of primary amine groups in both hydrogels can be estimated using the volume of HCl 0.1N added between these two endpoints (0.4 mL). As reported in Table 6, the amount of primary amine group was determined to be 0.7952 ± 0.0199 mmol/g for R4HG and 0.7576 ± 0.0379 mmol/g for R4HG-4I, which were very similar to those determined by Zu et al. for their 3D-Printable Hierarchical Nanogel-GelMA Composite Hydrogel System [51].

Considering data reported in Table 2 and that the cationic R4 contained in the amounts of R4HG and R4HG-4I utilized for titrations (50.3 mg and 52.8 mg, respectively) was 2.55 mg and 2.18 mg respectively, the results herein obtained were compared to those obtained previously in the titrations of resin R4 [23]. Since an NH_2_ quantity of 0.04 mmol per 2.55 mg and 2.18 mg of R4 coincided to 15.7 mmol and 18.3 mmol per gram of resin R4, respectively, against the determined value of 16.3 mmol/g, errors of −3.9% and +10.9%, respectively, were evidenced. The minor values obtained by titrating the already fully hydrated and swollen hydrogels R4HG with respect to the dry resin R4 could be justified supposing that the hydrogen bond interactions occurred during swelling between water, and the NH_3_ groups could have made the doublets of the nitrogen atoms less available to protonation, thus making such groups not titratable and providing an underestimation of the number of groups actually present. On the contrary, the major content in protonable nitrogen atoms in R4HG-4I compared to both R4HG and resin R4 can be justified by the presence of additional basic groups due to the contribution of **4I**. Collectively, since R4HG and R4HG-4I demonstrated to have amounts of ammonium groups per gram similar to those of R4 possessing potent antibacterial/bactericidal effects, they could be antibacterial as well.

#### 3.1.10. Rheological Studies

The rheological behavior of R4HG and of R4HG-4I was investigated by determining their apparent viscosity (η [Pa × s]) versus an applied shear rate (γ. [s^−1^]). Subsequently, we plotted η data vs. γ data, obtaining the graphs in Appendix A (R4HG) and Appendix A (R4HG-4I). Rheological experiments were performed on the prepared hydrogels to assess their rheological behavior, which is an important property for hydrogels finalized for biomedical applications [52].

The graphs evidenced that when the shear rate was close to zero, the viscosity of the hydrogel loaded with **4I** was significantly lower than that of R4HG. Anyway, in both cases η decreased rapidly for small increases of γ up to values of γ < 20, evidencing a non-Newtonian behavior. On the contrary, for values of γ > 20, η decreased very slowly becoming constant for γ > 40, demonstrating a behavior like Newtonian fluids.

Specifically, Newtonian fluids respect Newton’s law of viscosity. Accordingly, viscosity (η) does not depend on the shear rate (γ) and is constant. Shear stress (τ) of a Newtonian fluid is proportional to γ, and by plotting its τ vs. γ, a line with the constant slope (η) and intercept zero is obtained [53].

Hydrogels behaving as non-Newtonian fluids (η depending on γ) could be shear-thinning fluids, having values of the flow behavior index *n* < 1, when viscosity decreases with increasing γ. On the contrary, non-Newtonian fluids are shear-thickening dilatant fluids, having *n* > 1, when their viscosity increases with increasing γ. From the graph in Appendix A, for values of γ up to 100, R4HG and R4HG-4I behaved as shear-thinning fluids, with viscosity decreasing with increasing γ. This is an important trait of hydrogels that can be beneficial in their applications such as injection, drug delivery, and tissue engineering [52]. Anyway, among non-Newtonian shear thinning fluids, we can distinguish Bingham plastic fluids, whose viscosity is constant, the graph of τ vs. γ is linear, but intercept is >0, as well as pseudoplastic fluids and Bingham pseudoplastic fluids, whose viscosity is not constant and whose graphs of τ vs. γ are not linear and have intercept zero or >zero respectively [22,33,48].

Shear thinning is sometimes considered synonymous with pseudo-plastic behavior and is usually defined as excluding time-dependent effects, such as thixotropy [54,55,56].

Anyway, to assess precisely the rheological behavior of R4HG and R4HG-4I, we first determine the flow behavior index (*n*), and other parameters, and secondly, we obtained the plots of shear rate vs. shear stress. Briefly, we used the Cross Equation (14) present in the paper by Xie and Jin [57], and applying the hybrid mathematical method reported in the same study, we determined its four parameters *η*0, *η*∞, *α*, and *n*.
(14)$$\eta =\frac{\eta 0+\eta \infty \times \alpha \gamma^{n}\;}{\left(1+\alpha \gamma^{n}\right)}\;$$

In Equation (14), *γ* is the shear rate, η0 is the viscosity when the shear rate is close to zero, *η*∞ is the viscosity when the shear rate is infinite, *n* is the flow behavior index, and *α* is the consistency index. Table 7 reports the estimated parameters.

Secondly, to assess the goodness of the estimated parameters, we determined the viscosity values for both R4HG and R4HG-4I estimated by the Cross equation (CR viscosity) using such parameters and compared the plot of CR viscosity vs. *γ* with experimental plots of η vs. *γ* reported in Appendix A. From Appendix A, it can be seen that the CR viscosity and *γ* relationships obtained (fuchsia lines) were in reasonable agreement with the experimental results (purple lines).

Once we confirmed the goodness of the estimated parameters, we pursued our second goal consisting in obtaining the plot of shear stress (*τ*) vs. *γ*, to detect the exact rheological behaviours of our hydrogels among the possible shear thinning modes. To this end, we considered another form of the Cross Equation (15) containing three parameters (*α*, *n*, *τ*) [58] and the previous Equation (14), to construct a new Equation (16) obtained by comparing the denominators of Equations (14) and (15).
(15)$$\eta =\frac{\alpha }{\left(1+{\left(\frac{\alpha \gamma }{\tau }\right)}^{1-n}\right)}\;$$
(16)$$1+{\left(\frac{\alpha \gamma }{\tau }\right)}^{1-n}=1+\alpha \gamma^{n}\;$$

The *τ* values as a function of *γ* were obtained by solving Equation (16), and the related plots are reported in Appendix A.

According to Appendix A, both gels were Bingham pseudoplastic fluids not having constant η for shear rate values <50 s^−1^. Over these values, their η values were constant, thus behaving as plastic Bingham fluids characterized by the yield stress (*τ_o_*) parameter. The yield stress value (*τ_o_*), or tangential stress or critical strain point [59] reported in the last column of Table 7, in the case of pseudoplastic fluid corresponds to the intercept of the linear tendency lines in the linear tracts of their plots. For definition, the yield stress is the value of *τ* over which the pseudoplastic material begins to deform plastically. For values of *τ < τ_o_*, the hydrogel has a reversible elastic behavior, while for values of *τ > τ_o_* it has a plastic behavior characterized by the development of irreversible deformations. Low values of *τ_o_* as in our case (2.8 Pa and 25.71 Pa) are desired for hydrogels finalized to a topical application because they assure easy spreadability.

#### 3.1.11. Evaluation of **4I** in Vitro Release Profile

The in vitro release of **4I** from R4HG-4I and of **4I** from a **4I** suspension was evaluated spectroscopically (UV–vis) using the dialysis membrane method according to the literature [60]. Particularly, the experiments were performed in a simulated physiological medium (PBS, pH 7.4) at room temperature, and the measured concentrations of **4I** were used to compute the cumulative drug release percentage over time. The obtained data were plotted vs. time, obtaining the curves indicating the drug release profiles shown in Figure 9.

As observable, the release of **4I** from its suspension at the end of time considered was quantitative (99%). An important burst release within the first four hours leading to the discharge of the most part of **4I** (87%) was detected followed by a very small expulsion over the residual 68 h. On the contrary, although a burst release was observable also for R4HG-4I within the first 4 h leading to the discharged of 45% of contained **4I**, the release over the subsequent 24 h was significantly slower, followed by a plateau leading to a total release of 96%, thus evidencing the capability of R4HG-4I of allowing sustained and protracted release of **4I** vs. time. In this regard, the release profile of **4I** from R4HG-4I was significantly more controlled and sustained than that observed by Wang et al. for their fluorouracil (FU)-loaded pH-sensitive konjac glucomannan/sodium alginate (KGM/SA) and KGM/SA/graphene oxide (KGM/SA/GO) hydrogels [39].

##### Kinetic Studies

To better understand the kinetics and the main mechanisms which govern the release of **4I** from R4HG-4I, as previously described for other experiments, the data plotted in Figure 9 were fitted to the most common kinetic models [42,43,61,62]. The obtained dispersion graphs and the related linear regressions provided by Microsoft Excel software 365 using the OLS method are available in Appendix A. The highest value of their coefficients of determination (R^2^) was considered as the parameter to determine which model better fits the release data. The R^2^ values were reported in Appendix A (SM), and accordingly, the **4I** release from R4HG-4I best fitted with the Higuchi kinetic model (Appendix A), while that from the **4I** suspension with the Korsmeyer–Peppas model (Appendix A).

The Higuchi kinetics are expressed by Equation (17):(17)$$Qt=KH\times t^{n}\;$$
where *Qt* (variable y of the equation in Appendix A) is the amount of **4I** released at time *t*, *KH* is the Higuchi kinetic constant, *t* is time, and *n* is the release exponent, which in the Higuchi models is equal to 0.5 establishing that releases were ruled by Fickian diffusion as the releases of FU from pH-sensitive konjac glucomannan/sodium alginate (KGM/SA) and KGM/SA/graphene oxide (KGM/SA/GO) hydrogels reported by Wang [39]. The kinetic constant of the release of **4I** from R4HG-4I (*KH*) corresponded to the slope of the equation in Appendix A and was equal to 13.15. Concerning the release of **4I** from the **4I** suspension fitting the Korsmeyer–Peppas model end expressed by Equation (1), *n* (slope of the equation in Appendix A) was <0.5, thus establishing for a release ruled by Fickian diffusion as well. The kinetic constant of the release, corresponding to the intercept of the equation in Appendix A, was equal to 66.6, thus establishing that the release of **4I** from the suspension was 5 times faster than from R4HG-4I.

### 3.2. Biological Evaluations

#### 3.2.1. Cytotoxic Effects of **4I** and **4G** on MM Cells

To evaluate the cytotoxic effect of **4G** and **4I**, and to identify which one is the more suitable for further formulation studies, two human MM cell populations, one sensitive (MEOV NT) and the other resistant to PLX4032 (MEOV PLX-R), were exposed to both compounds, in a concentration range 0–100 µM for 24, 48, and 72 h. As reported in Figure 10 and Figure 11, both compounds exerted dose- and time-dependent cytotoxic effects on both cell populations. Moreover, at the same dose, **4I** resulted more cytotoxic than **4G** as demonstrated by IC_50_ values (Table 1).

In detail, 24-h treatment with 10 µM **4G** induced 17% of cell death of MEOV PLX-R (Figure 10b) while 20 µM **4G** caused 20% of cell death of MEOV NT (Figure 10a). However, the viability of both cell populations treated for 24 h with the maximal tested concentration (100 µM **4G**) was higher than 50%. Similar results were observed in MEOV PLX-R cells exposed to 100 µM **4G** for 48 h, while in MEOV NT cells a stronger cytotoxic effect was observed in comparison to 24-h treatment (Figure 10). In addition, cell viability of MEOV NT and MEOV PLX-R, treated for 72 h with 20 µM and 50 µM **4G,** respectively, was lower than 50%, while it decreased up to 23% and to 31%, respectively, in cells exposed to 100 µM **4G** (Figure 10). The analysis of **4I**-induced effects showed that the viability of MEOV NT and MEOV PLX treated for 24 h with 5 µM and 20 µM **4I,** respectively, was reduced by 15% (Figure 11), while the highest dose tested (100 µM) was able to induce a 35% decrease in both cell populations (Figure 11). The 48-h exposure to 0.5 µM **4I** caused 20% of cell death of MEOV PLX-R (Figure 11b) similarly to the effect observed in MEOV NT treated with 1 µM **4I** (17% of cell death; Figure 11a). Instead, after 72 h, **4I**-induced cytotoxic effect was similar in both cell populations (Figure 11). Overall, this result confirmed our previous data showing that **4G** and **4I** possess antiproliferative activity and that **4I** has greater activity than **4G** [24]. Interestingly, the data herein reported demonstrated that these compounds significantly reduced cells viability also in MM cells resistant to therapy with PLX, thus offering new adjuvant therapeutical options to treat melanoma. According to data previously reported [24], both compounds would exert cytotoxic effects on tumor cells affecting the cell-cycle phases with the appearance of polyploid cells, altering microtubules and targeting the tubulin system. The higher activity of **4I** respect to **4G** observed here could consequently depend on the different way of the two compounds interacting with the tubulin system [24]. Particularly, it seems that although the shorter chain of **4G** allows the molecule to fit into a small cleft formed by tubulin β and stathmin3, thus reinforcing the binding between the two proteins, molecule **4I** is positioned in a way that might prevent the correct binding of tubulin tyrosine ligase to the tubulin α chain [24]. Collectively, the cytotoxic effects of **4I** on both MEOV NT and on MEOV PLX-R were very close to the requirements of NCI (IC_50_ ≤ 10 µM) to be qualified as an active product in terms of in vitro antitumor activity. Already based on these findings, **4I** could be interesting to develop new agents to assist the cure of melanoma lesions at the beginning of the diagnosis in order to limit cancer growth and improve patients’ prognosis. More interestingly, according to the cytotoxic experiments carried out with PLX 0–20 µM on MEOV PLX-R melanoma cells at 72 h of exposure recently reported [25], when administered at the same concentrations (5 or 10 µM), **4I** was significantly more potent than PLX (Figure 12). As confirmation, the values of IC_50_ computed for both compounds in the concentration range 0–20 µM (Figure 12) were declared significantly different by a two-sample Student *t* test (95% and 99% confidence interval), the IC_50_ of **4I** being significantly lower than that of PLX.

As mentioned above, PLX is the main drug currently available to treat melanoma, whose use causes unfortunately the emergence of resistance in MM cells treated for a prolonged time, thus limiting its clinical application and efficiency. In this context, **4I** possessing an IC_50_ value 1.4-fold lower than that of PLX on resistant cells represents an interesting molecule worthy of further studies firstly finalized at enhancing its activity by proper formulation, as reported in this study. Due to these findings the subsequent biological analyses were focused on **4I**.

#### 3.2.2. Dose-Dependent Reactive Oxygen Species (ROS) Production in MEOV NT and MEOV PLXR Cells

Several chemotherapeutic drugs can exert cytotoxicity by increasing ROS production [32,63], and our recent study demonstrated that MEOV PLX-R cells display higher levels of glutathione (GSH), the most abundant intracellular antioxidant, compared to MEOV NT [25], allowing them to tolerate PLX. In this regard, to investigate if **4I** exerted cytotoxic effects also by this mode of action, the pro-oxidant action of **4I** was tested by analyzing the production of H_2_O_2_, the most long-lived ROS involved in redox modulation of signal transduction pathways [64,65] (Figure 13).

As shown in Figure 13, **4I** treatment stimulates H_2_O_2_ production by both cell lines in a time- and dose-dependent manner. In detail, 24-h exposure to 10 µM and 20 µM **4I** increased H_2_O_2_ production by 15% in MEOV NT and MEOV PLX-R (Figure 13), respectively, while the highest dose (100 µM) induced a 47% and 26% increase in MEOV NT and MEOV PLX-R, respectively, compared to untreated ones. At 48 h, a 20% increase of H_2_O_2_ was observed in MEOV NT and MEOV PLX-R cells treated with 1 µM and 0.5 µM **4I**, respectively (Figure 13). Instead, 100 µM **4I** increased H_2_O_2_ production by 127% (MEOV NT) and 47% (MEOV PLX) in comparison with untreated cells. In addition, 72-h treatment with 0.5 µM **4I** stimulated H_2_O_2_ generation by 20% in both cell populations, while 100 µM was able to increase peroxide production by 257% (MEOV NT) and 239% (MEOV PLX) compared to untreated cells (Figure 13). Collectively, both in MEOV NT and in MEOV PLX-R treated for 72 h with **4I** 20 µM, levels of ROS 3-fold and 2-fold higher than in control, respectively, were measured as reported in a study by Nambiar et al. for other cancer cells [66]. In the study, the authors demonstrated that the antiproliferative mechanism of an analogue of noscapine in different tumor cell lines depended on the alteration of tubulin system, in turn, dependent on production of a high level of ROS [66]. So, here we can speculate that, probably, the microtubule perturbation and cell-cycle phases alteration previously reported as the cause of tumor cells death when exposed to **4I** [24] could be triggered by the increased levels of ROS.

#### 3.2.3. Correlation between H_2_O_2_ Production and Cytotoxicity of **4I**

To better investigate the putative relationship between the cytotoxic and pro-oxidant effect of **4I**, the analyses of their correlation have been carried out by plotting the cell viability (%) vs. the ROS DCFC-positive cells (%) measured in the experiments carried out with **4I** on MEOV NT and MOV PLX-R melanoma cells, and the obtained dispersion graphs have been reported in Figure 14.

The values of R^2^ of the linear regressions obtained by the Ordinary Least Squares (OLS) method provided information about the correlation existing between the two parameters analyzed. According to the results reported in Figure 14a, an optimal correlation was shown in MEOV NT cells treated for 24 and 48 h, while a minor correlation was observed in cells treated for 72 h, suggesting that other factors, in addition to H_2_O_2_ production, are responsible for the cytotoxic effect of **4I** administered at the longest time. Instead, in MEOV PLX-R cells a good correlation was observed only at 72 h. These data suggest that in the drug-resistant cell population, **4I**, up to 48 h of treatment, exerted its cytotoxic action without inducing a marked increase in H_2_O_2_ production. Such an event was instead observed at 72 h of exposure, thus indicating that, at the longest time of treatment, the antioxidant defense of PLX-resistant cells is not able to contrast the OS induced by **4I**. Probably, at the shortest times of exposure, resistant cancer cells have an adequate antioxidant response able to guarantee their survival.

## 4. Conclusions

Hydrogels have a very wide domain of clinical applications, such as the treatment of traditional skin diseases, including melanoma, eye diseases, diabetes, brain and neuro diseases, as confirmed by numerous inspiring data from preclinical studies. Here, a hydrogel formulation containing an imidazo-pyrazole derivative (**4I**) has been prepared. To this end, we first screened the synthetized imidazo-pyrazoles **4G** and **4I** on two populations of melanoma cells (MEOV NT and MEOV PLX-R), finding **4I** more cytotoxic than **4G** on both cell lines and 1.4-fold more effective than PLX on MEOV PLX-R by an ROS-dependent mechanism. Then, aiming at improving further the cytotoxic activity of **4I** by a proper formulation strategy, it has been used to prepare a **4I**-based hydrogel (R4HG-4I) for future dermal administration. Interestingly, as gelling agent, we used synthetized antibacterial/bactericidal polystyrene-based cationic resin (R4) having high hydrophilicity, high-level porosity, and excellent swelling capabilities. Collectively, an empty R4-based hydrogel (R4HG) and a **4I**-enriched hydrogel (R4HG-4I) were obtained without using additives or additional gelling agents, thus avoiding undesirable and detrimental interactions of other chemicals with **4I** or skin irritations in a possible future use in vivo. The characterization of R4HG and R4HG-4I included ATR-FTIR analyses, water loss, and equilibrium swelling rate studies, determinations of their biodegradability in PBS over time, rheological and release experiments, and NH_3_^+^ group determinations, obtaining favorable results, fully supporting the suitability of R4HG-4I for topical uses. Due to these findings, the antibacterial effects of R4 and the ROS-dependent cytotoxicity of **4I**, higher than that of PLX on resistant melanoma, R4HG-4I can be considered for further experiments in a more realistic 3D model of melanoma, and upon favorable results could represent a starting point to develop new weapons to adjuvate PLX in melanoma cure. In detail, the novel topic hydrogel dosage forms which could derive by the optimization of the formulation developed here could be clinically used to reduce the dosage of PLX currently utilized, to limit the emergence of resistance, cancer growth and to improve patients’ prognosis. Although we are aware that the complete resection of tumor remains the main choice for melanoma management, our hypothesis is that R4HG-4I hydrogel could become an adjuvant therapy of second choice to treat melanoma lesions at the beginning of the diagnosis before the surgery or in a post-surgical step to reduce risk of infection, inflammation, and ulceration. The future perspective of this study currently in progress consists mainly in further biological experiments directly with the hydrogel in a more adequate 3D model of melanoma, such as melanoma spheroids embedded in extracellular matrix or organotypic skin reconstructs. Additionally, further studies including investigations on stability in different environments and in vivo conditions, as well as experiments of skin penetration, adhesive abilities, and topical drug release using the Franz diffusion cell approach, and cytotoxicity studies on normal cells will enrich the research presented here, which is only at the beginning.

## Data Availability

All data supporting the reported results are included in the present manuscript and in the associated Appendix A.

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
