# Peer review of "Antiproliferative Imidazo-Pyrazole-Based Hydrogel: A Promising Approach for the Development of New Treatments for PLX-Resistant Melanoma"

_pharmaceutics, 2023, doi:10.3390/pharmaceutics15102425_

Round 1

Reviewer 1 Report

In the manuscript titled “Antiproliferative Imidazo-Pyrazole-Based Hydrogel: a Promising Approach for the Development of New Treatments for PLX-Resistant Melanoma”, the authors have evaluated the efficiency of two dermal imidazo-pyrazoles (4G and 4I) based hydrogel formulations in treatment against two types of malignant melanoma cells - patient-isolated melanoma cells (MEOV NT) and Vemurafenib (PLX4032)-resistant (MEOV PLX-R). In the current manuscript, the authors designed and reported a thorough evaluation of both the chemistry (e.g., drug loading, entrapment efficacy, drug release kinetics, biodegradability, etc.) and biological aspects of the synthesized hydrogels. However, it is important to note that the individual components of the hydrogel, namely R4 and 4I/4G, had been previously studied for bactericidal effects and antiproliferative effects against several types of tumours including melanoma cells, which also has been duly cited by the authors in the current manuscript. Nevertheless, the reviewer recommends the following revisions to the manuscript before it can be accepted for publication. Please refer to the following comments/suggestions:

1. The introduction lacks information on the existing treatments/research against melanoma cells, the gap/weak spot in the same, and the author's contribution to it through this manuscript.

2. Please describe in more detail the objective of the current work and what has been done to achieve that in the introduction. In the current form of the manuscript, it is lacking.

3. Please define all the abbreviations (e.g. EBA, SPAN 85, APS, TMEDA, etc.) used and provide the chemical structures (3c, 4b, R1, etc.) even if it has been cited.

4. In the ‘Materials and Methods’ section, the temperature at which the water loss experiments were conducted has been reported at 37 °C while in the ‘Results and Discussion’ sections the temperature for the same experiments has been reported as 45 – 50 °C (Sec 2.2.5 and 3.1.7). Please check.

5. According to the reviewer, Figures S2 – S4 for R4 are redundant particularly because these have been published previously and seem identical to the previously published images. Consider removing them from the current supporting information and cite the reference in the current manuscript wherever needed.

6. Proofread for typos, incomplete sentences, and grammatical errors. E.g., pg. 2 lines 69 – 71, pg. 3 line 113, etc. 

Proofreading is required for typos, incomplete sentences, and grammatical errors.

Author Response

In the manuscript titled “Antiproliferative Imidazo-Pyrazole-Based Hydrogel: a Promising Approach for the Development of New Treatments for PLX-Resistant Melanoma”, the authors have evaluated the efficiency of two dermal imidazo-pyrazoles (4G and 4I) based hydrogel formulations in treatment against two types of malignant melanoma cells - patient-isolated melanoma cells (MEOV NT) and Vemurafenib (PLX4032)-resistant (MEOV PLX-R). In the current manuscript, the authors designed and reported a thorough evaluation of both the chemistry (e.g., drug loading, entrapment efficacy, drug release kinetics, biodegradability, etc.) and biological aspects of the synthesized hydrogels. However, it is important to note that the individual components of the hydrogel, namely R4 and 4I/4G, had been previously studied for bactericidal effects and antiproliferative effects against several types of tumours including melanoma cells, which also has been duly cited by the authors in the current manuscript. Nevertheless, the reviewer recommends the following revisions to the manuscript before it can be accepted for publication. Please refer to the following comments/suggestions:

  1. The introduction lacks information on the existing treatments/research against melanoma cells, the gap/weak spot in the same, and the author's contribution to it through this manuscript.

We thank the Reviewer for his/her comments which have enabled us to improve the part on current melanoma treatments and our contribute by the present work. Please, see lines 59-80 and 140-150.

  1. Please describe in more detail the objective of the current work and what has been done to achieve that in the introduction. In the current form of the manuscript, it is lacking.

We thank the Reviewer for his/her comments which have enabled us to better explain the scope of the present work. Please, see lines 80-84 and 124-127.

  1. Please define all the abbreviations (e.g. EBA, SPAN 85, APS, TMEDA, etc.) used and provide the chemical structures (3c, 4b, R1, etc.) even if it has been cited.

As asked, we have defined the abbreviations in the Scheme 1 caption, while the chemical structures of 3c, 4b, R1 and R4 have been provided as Figure S1 and S2 in the Supplementary Materials.

  1. In the ‘Materials and Methods’ section, the temperature at which the water loss experiments were conducted has been reported at 37 °C while in the ‘Results and Discussion’ sections the temperature for the same experiments has been reported as 45 – 50 °C (Sec 2.2.5 and 3.1.7). Please check.

We apologise to the Reviewer for our distraction. The correct data is 37°C. We have checked and corrected the wrong data (line 644-645).

  1. According to the reviewer, Figures S2 – S4 for R4 are redundant particularly because these have been published previously and seem identical to the previously published images. Consider removing them from the current supporting information and cite the reference in the current manuscript wherever needed.

As asked, Figure S2-S4 have been removed.

  1. Proofread for typos, incomplete sentences, and grammatical errors. E.g., pg. 2 lines 69 – 71, pg. 3 line 113, etc.

As asked, the missing term in lines 69-71 has been inserted (line 88, revised version), while the grammar of sentence starting in line 113 has been corrected. All manuscript has been checked to reduce all typos and grammatical errors. Additionally, it was revised by our colleague Prof. Deirdre Kantz, English teacher mother tongue working for the University of Genoa and Pavia.

Comments on the Quality of English Language

Proofreading is required for typos, incomplete sentences, and grammatical errors.

All manuscript has been checked to reduce all typos and grammatical errors. Additionally, it was revised by our colleague Prof. Deirdre Kantz, English teacher mother tongue working for the University of Genoa and Pavia.

Reviewer 2 Report

In this manuscript, authors reported the preparation of antiproliferative imidazo-pyrazole-based hydrogel as a potential dermal formulation against melanoma. The authors performed a series of the experiments for physicochemical characterization of the hydrogels. However, there are some major flaws in the design of this study. While developing a formulation for the dermal application the selection of the appropriate evaluation method that simulates the physiological condition is of utmost importance that is lacking in this manuscript. Therefore, I suggest major revisions for resubmission of this manuscript.

1. The level of English language is very poor, especially the writing style. The authors should thoroughly revise the language of the manuscript and use appropriate scientific terms and writing style  

2. What is the rationale (justification) for performing ATR-FTIR Spectroscopy, biodegradability, Water Loss Tests, rheological, and potentiometric titration experiments in the context of developing a dermal formulation?

3. The method section is long, please avoid unnecessary minor details.

4. The discussion of the results is very poor. The authors should discuss the significance of the result in the contexts of existing scientific literature on similar formulations.

6. The authors should perform the Scanning Electron Microscopic (SEM) analysis to study the morphology of the hydrogels.

7. The mechanical properties (Tensile strength and EAB) should be evaluated to investigate the suitability of the hydrogels for dermal application

8. Why biodegradability of the hydrogels was evaluated for 8 days. Is it possible to apply these hydrogels on the skin for 8 days?

9. Table 4: is Korsmeyer-Peppas model suitable to predict mechanism biodegradability (https://doi.org/10.1016/0378-5173(83)90064-9 )?

10. Swelling Rate Experiment: Does the experimental conditions for this experiment simulate the dermal condition (skin is usually dried or mildly moist)? Why the swelling rate of R4HG-4I is higher than R4HG (Figure 6)? Please also add the error bars in Figure 6.

11. Drug release experiment: It is better to use the Fraz-diffusion cell for dermal formulation.

12. Authors should perform live dead assay and cellular uptake studies in cell line

The level of English language is very poor, especially the writing style. The authors should thoroughly revise the language of the manuscript and use appropriate scientific terms and writing style  

Author Response

In this manuscript, authors reported the preparation of antiproliferative imidazo-pyrazole-based hydrogel as a potential dermal formulation against melanoma. The authors performed a series of the experiments for physicochemical characterization of the hydrogels. However, there are some major flaws in the design of this study. While developing a formulation for the dermal application the selection of the appropriate evaluation method that simulates the physiological condition is of utmost importance that is lacking in this manuscript. Therefore, I suggest major revisions for resubmission of this manuscript.

1. The level of English language is very poor, especially the writing style. The authors should thoroughly revise the language of the manuscript and use appropriate scientific terms and writing style  

All manuscript has been checked to reduce all typos and grammatical errors. Additionally, it was revised by our colleague Prof. Deirdre Kantz, English teacher mother tongue working for the University of Genoa and Pavia, where she teaches scientific English in the degree courses in Pharmacy and Pharmaceutical Chemistry and Technology.

  1. What is the rationale (justification) for performing ATR-FTIR Spectroscopy, biodegradability, Water Loss Tests, rheological, and potentiometric titration experiments in the context of developing a dermal formulation?

As reported in the text (lines 501-504) and as reported by other authors (https://doi.org/10.1038/s41427-021-00354-4), ATR-FTIR spectroscopy analyses were performed to investigate the chemical structure and the composition of the prepared hydrogel. In fact, ATR-FTIR spectra provided essential information about the functional groups present in the gel thus assuring the correct entrapment of 4I in the 3D network formed by resin R4 and the absence of side products.

Biodegradability of the developed hydrogel was assessed because it is an essential property of hydrogels finalized for biomedical applications (https://doi.org/10.1016/j.addr.2023.115028). Specification on this question have been included in the main text (lines 593-594).

We thought to perform water-loss experiments at 37°C to evaluate the capability of the prepared hydrogels to retain water when air exposed and therefore a possible exudate in the melanoma ulcerations, and the time necessary for the gel to reach full dryness thus losing its spread ability. These specifications have been added in the main text (646-649).

Rheological experiments were performed on the prepared hydrogels to assess their rheological behaviour (shear-thickening or shear-thinning behaviour), which are important properties of hydrogels finalized to biomedical applications as topical formulations. Particularly, a shear thinning behaviour, as that demonstrated by R4HG and R4HG-4I reported here, can be beneficial in their applications such as injection, drug delivery, and tissue engineering (https://doi.org/10.3390%2Fgels7040255). These specifications have been added in the main text (lines 816-818 and 833-835).

Potentiometric titrations of R4HG and R4HG-4I were performed to determine the absolute amount of cationic ammonium groups deriving by R4 present in the prepared hydrogels. This data is important for the antibacterial properties of cationic materials and to forecast those of the prepared hydrogels. Additionally, a high number of cationic groups could improve the cytotoxicity of 4I by allowing electrostatic interactions with the negatively charged surface of tumour cells, thus damaging cells membrane and triggering cells death (https://doi.org/10.3390/ijms23116108). These specifications have been added in the main text (lines 759-765).

  1. The method section is long, please avoid unnecessary minor details.

As asked, the methods section was shortened.

4. The discussion of the results is very poor. The authors should discuss the significance of the result in the contexts of existing scientific literature on similar formulations.

The discussion of the results has been improved by inserting in all section, where not already reported, comparisons of the obtained result with those existing in scientific literature on similar formulations. Please see lines 473-476, 491-499, 577-590, 610-613, 788-791, 896-899 and 912-917.

  1. The authors should perform the Scanning Electron Microscopic (SEM) analysis to study the morphology of the hydrogels.

As asked the SEM analysis was carried out and results were included in the paper. Please, see lines 208-210 and 491-495.

  1. The mechanical properties (Tensile strength and EAB) should be evaluated to investigate the suitability of the hydrogels for dermal application

We thank a lot the Reviewer for his/her suggestion, but the suitability of the prepared hydrogel for dermal application was assessed as also reported in literature by determining the flow curves of viscosity vs. shear rate and then the curves of shear stress vs. shear rate, since they provide indication on how hydrogels flow. Flow curves show if hydrogels are shear thickening or shear thinning fluids, where shear thinning fluid are desirable for hydrogel formulations to be applied topically. We kindly ask the Reviewer to be satisfied with the rheological experiments performed by us.

 Why biodegradability of the hydrogels was evaluated for 8 days. Is it possible to apply these hydrogels on the skin for 8 days?

According to the reported methods to evaluate the biodegradability of a hydrogel by assessing its mass loss, the experiment must be performed until a constant weight (equilibrium) is reached. Particularly, we evaluated the biodegradability of R4HG and R4HG-4I according to the literature cited in the text [Ref 46, revised version]. Additionally, studies exist in literature where degradability of hydrogels was evaluated for weeks.

9. Table 4: is Korsmeyer-Peppas model suitable to predict mechanism biodegradability (https://doi.org/10.1016/0378-5173(83)90064-9 )?

Yes. According to literature [Ref. 47 revised version] and as reported in the text, it is possible to get the mechanism by which the release of something (mass in our case) from a polymer system is governed, on the base of the higher R2 of the linear regressions of the dispersion graphs obtained fitting the different mathematical model to water loss (%) data. Therefore, the results reported in Table 4 indicated that the mass loss of the hydrogels was better described by the Korsmeyer Peppas kinetic model.

  1. Swelling Rate Experiment: Does the experimental conditions for this experiment simulate the dermal condition (skin is usually dried or mildly moist)?

The swelling rate experiments serve to investigate how much water over time a dried gel is capable to absorb and retain and the experimental conditions do not need to simulate the dermal conditions. When dermal administered, the hydrogel is already swollen at its maxima EDS and EWC and do not need of skin moisture to swell. Anyway, experiments were performed as described in literature (Zhang, K.; Feng, W.; Jin, C. Protocol efficiently measuring the swelling rate of hydrogels. MethodsX 2020, 7, 100779. doi:10.1016/j.mex.2019.100779).

Why the swelling rate of R4HG-4I is higher than R4HG (Figure 6)?

As reported (Abasalizadeh, F., Moghaddam, S.V., Alizadeh, E. et al. Alginate-based hydrogels as drug delivery vehicles in cancer treatment and their applications in wound dressing and 3D bioprinting. J Biol Eng 14, 8 (2020). https://doi.org/10.1186/s13036-020-0227-7) the composition of hydrogels strongly affects their swelling behaviour. In our case, the presence of 4I in the 3D network of R4HG, having several heteroatoms capable to give hydrogen bonds with water, improved the capability of the hydrogel to interact with water, to absorb it and its swelling rate increased. This explanation has been added in the main text (lines 705-709).

Please also add the error bars in Figure 6.

As asked, error bars have been added in the old Figure 6, now Figure 7.

  1. Drug release experiment: It is better to use the Fraz-diffusion cell for dermal formulation.

We agree with the Reviewer, but this is only the first step of our research on this imidazo-pyrazole loaded hydrogel. More in deep release studies using the Franz-diffusion cell will be performed upon promising results on a more realistic 3D model of melanoma.

  1. Authors should perform live dead assay and cellular uptake studies in cell line

We thank the Reviewer for his/her suggestion, but as underlined in the previous point, this is only the first step of our research on this imidazo-pyrazole loaded hydrogel, and the experiments suggested are out of scope of this work. More in deep biological investigations will be performed upon promising results on a more realistic 3D model of melanoma.

Comments on the Quality of English Language

The level of English language is very poor, especially the writing style. The authors should thoroughly revise the language of the manuscript and use appropriate scientific terms and writing style  

All manuscript has been checked to reduce all typos and grammatical errors. Additionally, it was revised by our colleague Prof. Deirdre Kantz, English teacher mother tongue working for the University of Genoa and Pavia, where she teaches scientific English in the degree courses in Pharmacy and Pharmaceutical Chemistry and Technology.

Reviewer 3 Report

1. In the Abstract it is advisable to use the full names of the studied compounds instead of 4I and 4G.

2. I suppose "inverse" is more proper instead of "reverse" for the suspension system.

3. Please, recheck all the References entries and prepare them in accordance to "Guide for Authors".

4. Please, recheck for typos, grammatical errors etc.

Please, recheck for typos, grammatical errors etc.

Author Response

Comments and Suggestions for Authors

  1. In the Abstract it is advisable to use the full names of the studied compounds instead of 4I and 4G.

As asked the full chemical names of 4I and 4G have been included in the abstract (lines 16-18).

  1. I suppose "inverse" is more proper instead of "reverse" for the suspension system.

We make kindly note to the Reviewer that the correct name of the polymerization method is just “reverse-phase suspension copolymerization technique”. Please see https://doi.org/10.1002/masy.19920540141. Anyway, we thank the Reviewer for his/her comment.

  1. Please, recheck all the References entries and prepare them in accordance to "Guide for Authors".

Done

  1. Please, recheck for typos, grammatical errors etc.

Done

Comments on the Quality of English Language

Please, recheck for typos, grammatical errors etc.

All manuscript has been checked to reduce all typos and grammatical errors. Additionally, it was revised by our colleague Prof. Deirdre Kantz, English teacher mother tongue working for the University of Genoa and Pavia, where she teaches scientific English in the degree courses in Pharmacy and Pharmaceutical Chemistry and Technology.

Round 2

Reviewer 1 Report

Thank you authors for addressing the comments/suggestions in the revised version of the manuscript. There are a few minor grammatical errors (punctuation, tense, adverb, etc.) in the current version, which require your attention. The reviewer recommends the publication of the manuscript.

There are a few minor grammatical errors (punctuation, tense, adverb, etc.) in the current version, which require the author's attention.